# Southern Africa crustal anisotropy reveals coupled crust-mantle evolution for over 2 billion years

H. Thybo [1,2,3]*, M. Youssof[4] & I.M. Artemieva [5]

The long-term stability of Precambrian continental lithosphere depends on the rheology of the lithospheric mantle as well as the coupling between crust and mantle lithosphere, which may be inferred by seismic anisotropy. Anisotropy has never been detected in cratonic crust. Anisotropy in southern Africa, detected by the seismological SKS-splitting method, usually is attributed to the mantle due to asthenospheric flow or frozen-in features of the lithosphere. However, SKS-splitting cannot distinguish between anisotropy in the crust and the mantle. We observe strong seismic anisotropy in the crust of southern African cratons by Receiver Function analysis. Fast axes are uniform within tectonic units and parallel to SKS axes, orogenic strike in the Limpopo and Cape fold belts, and the strike of major dyke swarms. Parallel fast axes in the crust and mantle indicate coupled crust-mantle evolution for more than 2 billion years with implications for strong rheology of the lithosphere.

[1] School of Earth Sciences, China University of Geosciences, Wuhan, China. [2] Eurasia Institute of Earth Sciences, Istanbul Technical University, Istanbul, Turkey. [3] Centre for Earth Evolution and Dynamics (CEED), University of Oslo, Oslo, Norway. [4] King Abdullah University of Science and Technology (KAUST), Thuwal, Saudi Arabia. [5] Stanford University, Palo Alto, CA, USA. *email: thybo@geo.uio.no

An enigmatic feature of Precambrian continental lithosphere is its long-term stability[1] after formation and amalgamation more than 2 billion years ago. The stability of the lithosphere system of crust and mantle depends primarily on the rheology of the lithospheric mantle[2,3] as the part of the lithosphere directly affected by mantle flow, and also on the degree of coupling between the crust and mantle since cratonisation.

The early-to-late Archaean cratons in southern Africa constitute a natural laboratory for studies of the formation of cratons and their retention for billions of years. Surface geologic mapping and chronology, together with results from isotope studies on xenoliths, show that the cratonic crust of southern Africa formed and stabilised in the Archaean[4] and that it was reworked by a series of Proterozoic and Phanerozoic tectono-magmatic events[5].

The depth extent of the southern African cratons has been studied by several methods that indicate substantially different estimates for the thickness of the lithosphere. Seismic body-wave tomography, by assuming an isotropic mantle, indicates lithosphere thicknesses of up to 250 km by application of ray-theory methods[6] and up to 350 km by application of finite-frequency methods[7], respectively. Seismic surface wave tomography indicates that the lithosphere thickness is around 175 km when possible anisotropy is not taken into account[8]. Surface wave inversion taking anisotropy into account indicates that the upper crust is highly anisotropic in the Limpopo Belt, and that both the mantle lithosphere and the asthenosphere are anisotropic with different directions of the fast axes[9].

Anisotropy is direction-dependent seismic velocity. It is most often detected by the so-called SKS-splitting method that identifies the accumulated anisotropy between the Earth's core and the surface by measuring the travel-time difference between the horizontally (SH) and vertically (SV) polarised waves. This method cannot provide depth control, and it is mostly assumed that the main anisotropy resides in the lithospheric mantle or the asthenosphere[10]. The causes of anisotropy are alternating isotropic layers with different elastic properties[11], alignment of joints or microcracks filled with water or melts[12–14], foliated metamorphic rocks[15] and lattice preferred orientation (LPO) of anisotropic minerals. Olivine minerals are highly anisotropic and develop preferred lattice orientation as response to finite strain, such that mantle peridotite often exhibits strong anisotropy[16–19].

Seismic studies provide insight into the structure of the cratonic mantle[20–23] but also introduce some controversy regarding the evolution of the lithosphere. Earlier seismic studies infer deformation below the lithosphere by mantle flow with the fast direction of seismic anisotropy being parallel to present plate motion[23] or anisotropy frozen into the lithospheric mantle[21,24]. Earlier observations of strong crustal anisotropy have been restricted to tectonically young areas[25–28].

We present the first observation of strong anisotropy in cratonic crust from analysis of seismic data from southern Africa. We analyse the nature of the evolved cratonic lithosphere by evaluating seismic anisotropy in the crust and correlating it with crustal terranes[29], major dyke swarms[30] and anisotropy determined by SKS splitting[21]. We find that the fast axes of the anisotropy are homogeneous within the main tectonic units, and that they are parallel to the fast axes determined by SKS analysis, to the orogenic strike in the Limpopo and Cape fold belts, and to the strike of major dyke swarms. Parallel fast axes in the crust and mantle indicate that the crust and mantle have been coupled since the amalgamation 2 billion years ago, which requires strong rheology of the whole lithosphere.

## Results

**Seismic observations of anisotropy.** Analysis of SKS wave motion is a frequently used technique for determination of azimuthal anisotropy, yielding high lateral resolution of the total anisotropy along ray paths but without depth control. For analysis of anisotropy in the crust, we instead apply the receiver function (RF) technique for determination of P- to S-wave (Ps) conversions at interfaces based on the high-quality data of the Southern African Seismic Experiment (SASE) (Fig. 1). In contrast to earlier assumptions of weak crustal anisotropy (less than a quarter of the total anisotropy along the whole ray path for core phases) for the earlier analysed half of the SASE stations[21], our results show that the crustal contribution to the total anisotropy is significant at all SASE stations. It is on average 30% (reaching >50% at some stations) of the total SKS splitting, and the direction of the fast axis ($\varphi$) for the crustal anisotropy is uniform within each tectonic block (Fig. 1).

The current analysis is based on 6198 RFs from 220 teleseismic events (Mw ≥ 5.5) with broad distribution of azimuth and distance (Fig. 2). The SASE seismic data[21] were acquired by 55 broadband instruments deployed at 82 locations during the period from April 1997 to July 1999. Average spacing between stations is ca. 100 km in a ca. 2000-km-long SW–NE striking corridor across the Kalahari Craton (Fig. 1). We calculate the RFs by using the LQT method[31,32], which suppresses the almost vertical P-wave motion energy by decomposing the seismic wavefield into the L, Q, and T components, i.e., the compressional wave (P), the vertically polarised S-wave (SV) and the horizontally polarised S-wave (SH) (cf. Methods section).

Observation of energetic transverse ($P_{SH}$-RF) phases and different travel times of radial ($P_{SV}$) and transverse ($P_{SH}$) RFs (Fig. 3) shows that earlier assumptions of an isotropic crust[21,33,34] cannot explain the seismic data. In the following, we use the terms $P_{SV}$-RF phases for the radial component in the plane of the back azimuth and the vertical direction and $P_{SH}$-RF phases for the transverse-component phases as inherent in the LQT method and the software used for calculation of synthetic RFs[35]. Our calculated RFs show azimuthally dependent features within the first 5 s, including periodic amplitude variation of the Ps phase, polarity reversals and Ps delay undulation (Fig. 4), which are the discriminators for crustal anisotropy[36]. Stacks of the RFs in back-azimuth bins from all stations within individual tectonic units of the cratons[29] demonstrate that the RF observations are related to crustal anisotropy and not determined by local structure at the individual stations (Supplementary Figs. 1–4).

**Quantifying anisotropy by modelling.** Azimuthal anisotropy is indicated by the time delay ($\delta t$) between the radial (SV) and transverse (SH) converted waves[13]. The average $\delta t$ between $P_{SV}$ and $P_{SH}$ generated at the base of the lower cratonic crust (Fig. 3) is 0.19 s for the Archaean cratonic areas (smallest in the Limpopo Belt, 0.16 s) and 0.34 s in the Cape and Namaqua–Natal post-Archaean fold belts (Supplementary Table 1). These delay times originate in the crust and amount to at least 30–55% of the S-wave splitting of 0.62 s measured on SKS phases in the Kaapvaal and Zimbabwe cratons[21]. This is remarkable as the crust is only 40 km thick in comparison with the >160-km-thick lithospheric mantle. This is the first observation ever made of strong anisotropy in Archaean cratonic crust. Crustal anisotropy is usually considered negligible in interpretations of the observed shear-wave splitting of SKS phases for determination of anisotropy in the upper mantle[21,23]. Our new observation suggests that this assumption may not be true, not even in cratonic regions.

All sections show phase reversal of the transverse $P_{SH}$-RF component with a back-azimuthal period of $2\pi$ (Fig. 4) as

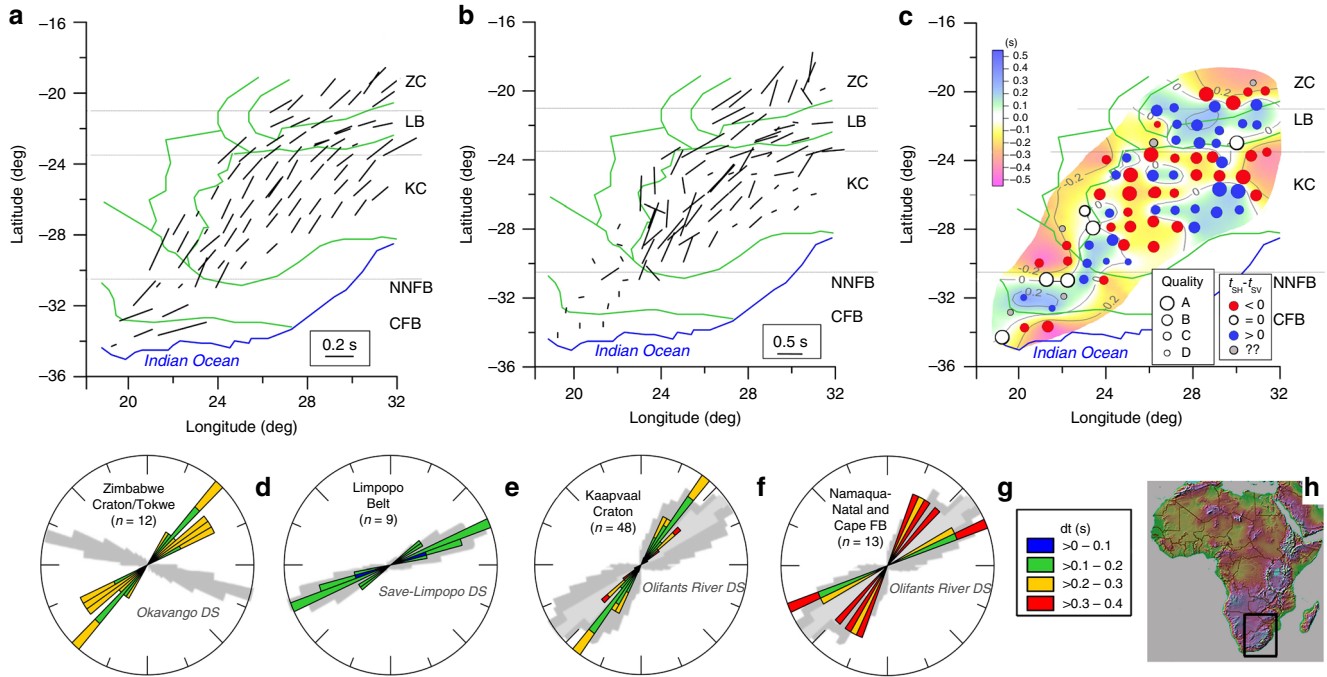

**Fig. 1** Anisotropy variation in southern Africa: fast polarisation direction for **a** crustal anisotropy from from surface to Moho based on RFs, **b** total anisotropy determined from SKS splitting[21] (another scale than in (**a**)), **c** sign of difference in arrival times of transverse $S_H$ and radial $S_V$ phases (d*t*) where size of station symbols indicates quality of the determined difference. *Abbreviations*: ZC Zimbabwe Craton, LB Limpopo Belt, KC Kalahari Craton, NNFB Namaqua–Natal Fold Belt, CFB Cape Fold Belt. **d–g** Rose diagrams of observed fast directions for all analysed stations in the various tectonic provinces, colour coded for delay time d*t*, superimposed on directions of individual dykes (grey shading[30]) in major dyke swarms (DS) for four main areas; fold belt is abbreviated FB. **h** Map shows the study area. All the 82 stations from the SASE experiment are analysed for crustal anisotropy.

predicted for an anisotropic medium[36,37]. Stacking is required to accurately determine the time delay δ*t* but will lead to insignificant energy in the stacked trace due to the polarity change if applied directly to the RFs. We therefore stack only positive polarity radial $P_{SV}$- and transverse $P_{SH}$-RFs, which provides sufficient signal/noise ratio to identify the phases instead of attempting to make phase reversals before stack, because such correction may affect the results. This approach is justified by the fact that we stack signals with a period of about 1.5 s and the determined time delay δ*t* is generally <0.5 s, which makes the widening effect insignificant on the stacked signals with slightly different travel times. The small observed undulations in arrival time of the phases due to anisotropy are insignificant compared with the period of the waveform, and this allows for direct stacking without prior alignment. Therefore, by this approach, we obtain a robust estimate of the minimum delay time caused by crustal anisotropy.

Our RFs (Fig. 3) include two converted phases from the top of the lower crust (labelled LC) and from the Moho (labelled M) that both show evidence for anisotropy. The azimuthal amplitude variation of these phases is sinusoidal with a phase difference of π, which we model with synthetic RFs (Fig. 4, Supplementary Figs. 1–4) calculated by a ray-based algorithm[35] that incorporates both dipping anisotropy and independently dipping interfaces. By visual inspection and comparison, we conclude that average anisotropic velocity models (Table 1) explain the main features of the observed RFs from Kaapvaal and Limpopo, although the determined parameters are non-unique due to uncertainties and nonlinearity. The signal-to-noise ratio is too low to permit application of quantitative measures of the correlation between synthetic and observed seismograms as also found in studies of tectonically young regions[25–28]. The modelled relative velocity between the fast and slow axes is very high (10%) in Kaapvaal lower crust and 7% in the whole crust of the Limpopo Belt.

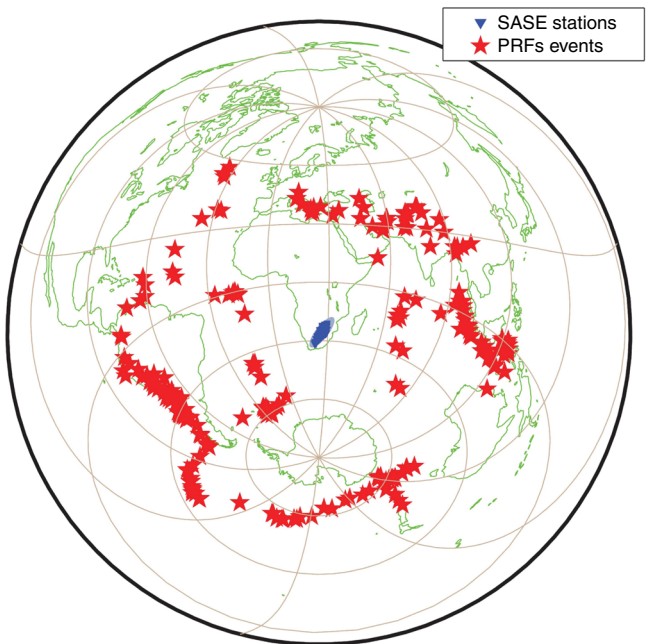

**Fig. 2** Event distribution for the calculation of receiver functions. Red stars: seismic events, blue area: region covered with seismic stations during the SASE experiment. The events cover all back-azimuth directions with a broad distribution of epicentral distances.

For the Kaapvaal and eastern Zimbabwe cratons, the transverse $P_{SH}$-RF (Fig. 4a) identifies a polarity change at a back azimuth of 40° (and 220°) ± 20° for the strong phase M from the Moho and at 100° ± 20° for the weak phase LC from the top of the lower crust (Fig. 4a), which indicates the directions of the fast polarisation axes

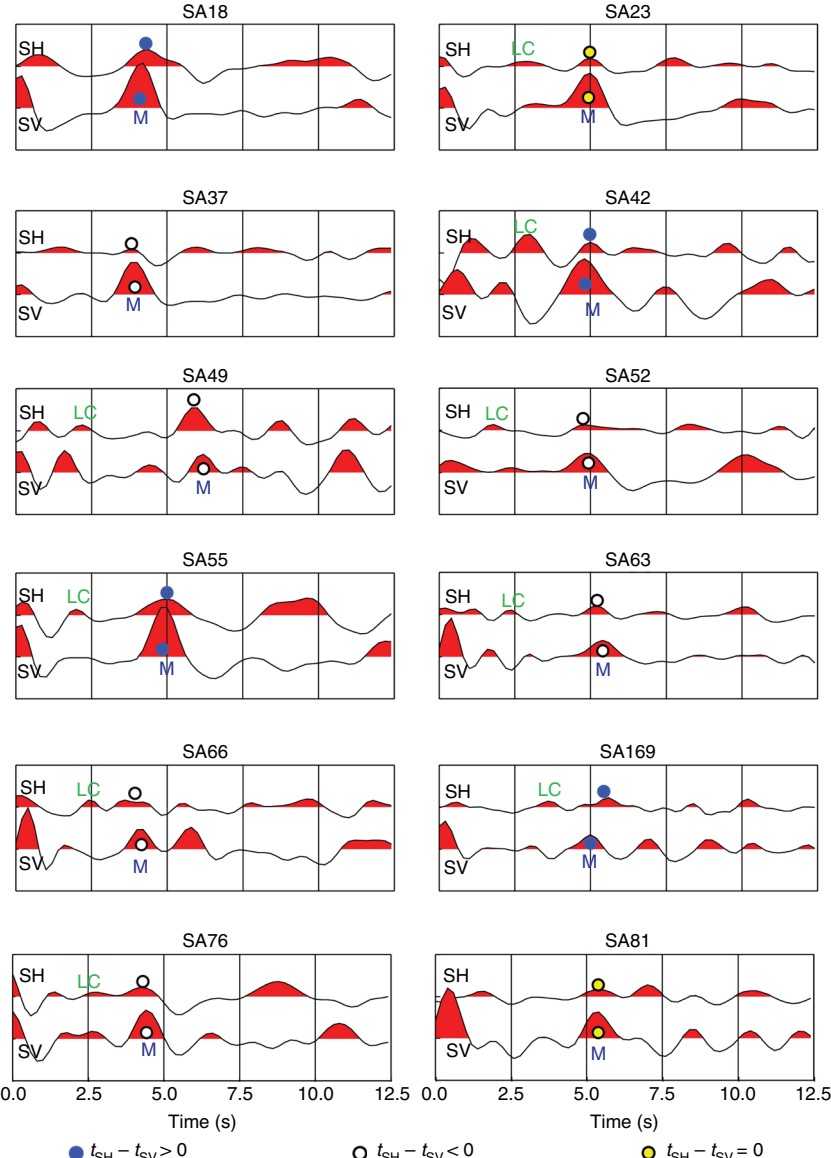

**Fig. 3** Stacked transverse (P$_{SH}$) and radial (P$_{SV}$) receiver functions used to calculate the amplitude (δt) of crustal anisotropy (cf. Supplementary Table 1). Only positive amplitudes were stacked; otherwise, the resulting stack would be close to zero amplitude due to the polarity changes incurred by anisotropy. The examples are from Kaapvaal Craton (SA18, 37, 42, 49, 52, 63), Limpopo Belt (SA55, 169), Zimbabwe Craton (SA66, 76), Kheis Belt (SA23) and Namaqua–Natal Belt (SA81). M and LC denote conversions from the Moho and top of the lower crust, respectively. Notice the change in relative arrival times of the transverse P$_{SH}$- and the radial P$_{SV}$-RFs between the two cratons and the Limpopo Belt.

(φ) in the lower and upper crust. The 2π periodicity of the LC and M phases and the π/2 phase lag between the radial P$_{SV}$ and transverse P$_{SH}$ components of the LC phase show that the symmetry axis of the anisotropy cannot be horizontal[36–38]. In western Zimbabwe Craton, the direction of the fast polarisation axes is less uniform between 50 and 70° (Supplementary Table 1). The direction of the fast axes in the Limpopo Belt and the Cape Fold Belt (Fig. 4b) is 70° ± 20° for the crust, which is distinctively different from the Kaapvaal and Zimbabwe cratons (Fig. 1a).

## Discussion

It is characteristic for the Limpopo Belt and western Zimbabwe Craton that the transverse P$_{SH}$ component arrives earlier than the radial P$_{SV}$ component, whereas the opposite is the case in other areas, although with some scatter (Fig. 1c, Supplementary Table 1). This observation can be explained by the dip of the fast axis, which we model to be 50° from vertical in most of Kaapvaal

and 35° in the Limpopo and Cape fold belts (Table 1). Sensitivity analysis (Supplementary Fig. 5) indicates that the uncertainties of the determined anisotropy are around ±4% for the strength and ±10° for the plunge angle around the chosen best parameters by assuming a standard deviation of δt = ± 0.1 s. However, the strong nonlinearity of the relations between strength, plunge angle and δt makes direct estimation of uncertainties non-unique, cf. Supplementary Figs. 5 and 6.

The fast directions for the crustal and the SKS anisotropy are generally parallel (Figs. 1a, b and 5) although the anisotropy is much less coherent in the SKS-splitting results than in our crustal results from RF (Figs. 1, 5, 6 and Supplementary Fig. 7), particularly in the Cape and Namaqua–Natal fold belts and western Kaapvaal. A study of shear-wave splitting based on seismological data from a dense array around Kimberley has shown that mantle anisotropy may vary at shorter wavelength than the spacing between the stations of the SASE array[39].

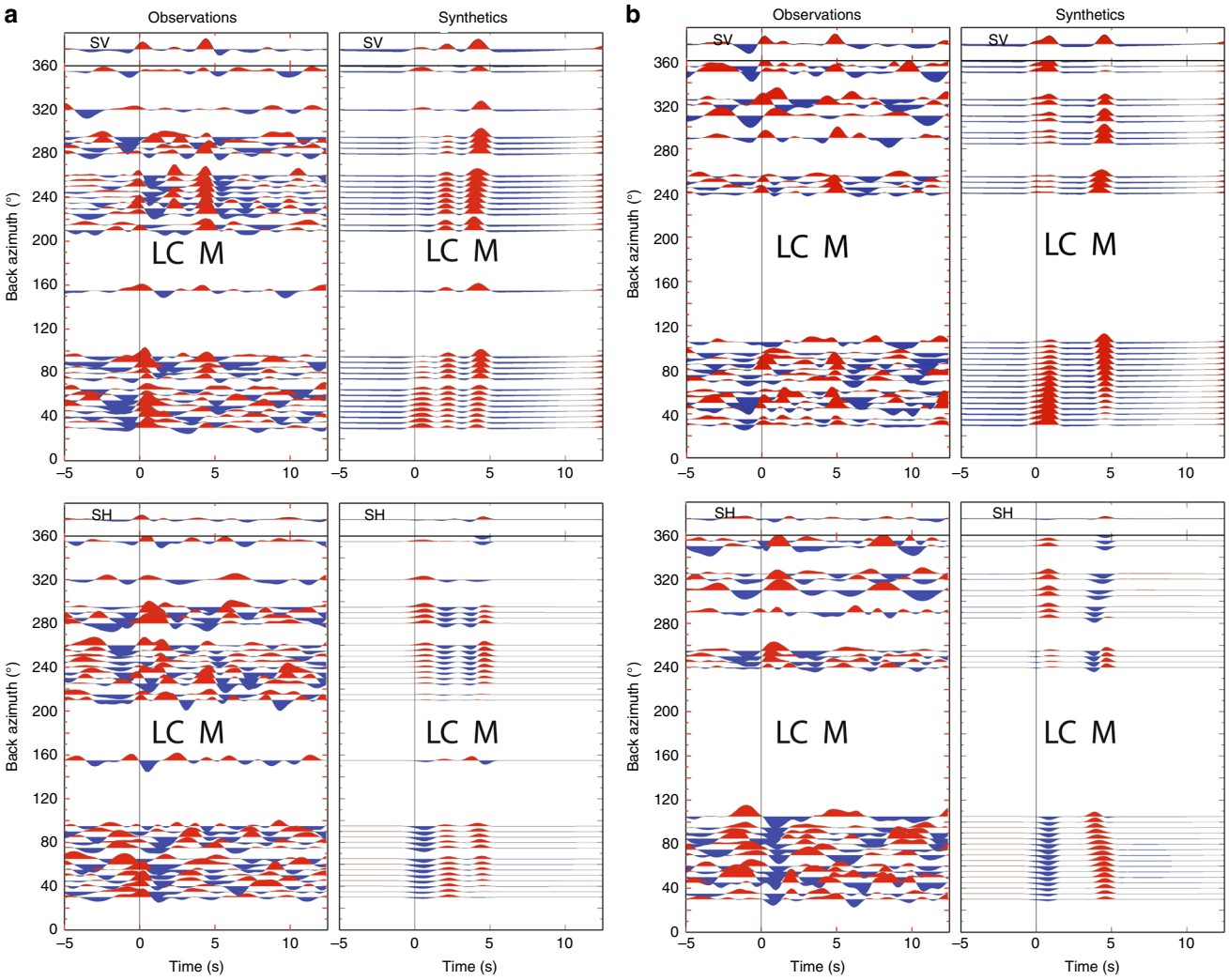

**Fig. 4** Transverse P$_{SH}$ and radial P$_{SV}$ receiver functions versus back azimuth for the Kaapvaal Craton (**a**—station SA25) and Limpopo Belt (**b**—station SA55) and synthetic receiver functions calculated for the models in Table 1 (stacked in 5° bins). LC—top lower crust conversion, M—Moho conversion. Supplementary Figs. 1–4 show similar results based on stacked RFs for all stations in the corresponding tectonic provinces.

A recent study of primarily upper mantle anisotropy, based on analysis of surface waves, indicates similar direction of the fast axis in the asthenosphere and the direction measured by SKS-splitting analysis, whereas the results indicate substantial deviation for the lithospheric anisotropy[24]. The surface waves have some depth resolution and the authors observe parallel fast axes in the asthenosphere, which are parallel to the plate motion. The observed strong anisotropy in both the lithospheric and asthenospheric mantle generally has parallel fast axes, except for those in the Zimbabwe craton and to some degree in the Limpopo Belt, where surface waves indicate a more east–west-directed fast axis in the asthenosphere than the SKS results[9].

Although the SKS direction may be affected by the crustal anisotropy by rotating the splitting fast axis along the ray paths[40], the observed coherence is surprising because the SKS fast-propagation direction may not necessarily be the same in all depth intervals of the crust, and lithospheric and sublithospheric mantle[9,23,24]. We interpret the similar anisotropy directions in the Kaapvaal and Zimbabwe cratons that geologically developed independently until 2.7 Ga, as an indication that the anisotropy was frozen into the lithosphere during the collision between these cratons at ca. 2.7 Ga when the Limpopo belt was deformed such that it today has different anisotropy fast axes. The coherence of the crustal and the SKS anisotropy directions in the two cratons may further provide indication for this interpretation, although the SKS-splitting results are affected by the crustal anisotropy[40]. This conclusion is further supported by the observations of a surprising coherence between the crustal anisotropy directions in the cratons and the orientations of major dyke swarms except for the Zimbabwe Craton (Fig. 1), and between the crustal and SKS anisotropy directions in the intervening Limpopo Belt where the fast directions are subparallel to the strike of the collisional belt (Figs. 5, 6).

We suggest that lattice-preferred orientation (LPO), alignment of anisotropic minerals and structural anisotropy (SPO) associated with dyke intrusions may account for the modelled S-wave perturbation ($\delta Vs\%$, Table 1) in the crust. As such, the causes for crustal anisotropy may be more complex than for mantle anisotropy, which is believed to primarily originate from LPO of olivine aggregates[10,19]. The azimuthal anisotropy observed in the brittle upper crust may be related to upper crustal fine layering or aligned cracks (confined to depths <5–10 km), but this mechanism cannot explain our observations entirely because such anisotropy usually is weak[41] (up to 5%). However, the fast axes are also parallel to the large dyke swarms in the Limpopo Belt, Kaapvaal Craton and the Namaqua–Natal and Cape fold

| Thickness km | Density, g cm$^{-3}$ | Vp, km s$^{-1}$ | Vs, km s$^{-1}$ | S-anisotropy % | Fast direction degree | Plunge angle degree | Layer strike degree | Layer dip degree |
|---|---|---|---|---|---|---|---|---|
| *Kaapvaal Craton* | | | | | | | | |
| 0.7 | 2.6 | 5.85 | 3.32 | 0 | 0 | 0 | 0 | 0 |
| 5.1 | 2.7 | 6.15 | 3.48 | 1 | 55 | 20 | 0 | 0 |
| 11.2 | 2.8 | 6.45 | 3.66 | 2 | 35 | 50 | 0 | 0 |
| 20.5 | 2.86 | 6.73 | 3.82 | 10 | 35 | 50 | 0 | 0 |
| 0 | 3.3 | 8.2 | 4.65 | 0 | 0 | 0 | 0 | 0 |
| *Limpopo Belt* | | | | | | | | |
| 0.5 | 2.6 | 5.9 | 3.45 | 0 | 0 | 0 | 0 | 0 |
| 9 | 2.79 | 6.3 | 3.7 | 4 | 75 | 5 | 60 | 2 |
| 35 | 2.86 | 6.8 | 4 | 7 | 75 | 33 | 60 | 2 |
| 0 | 3.3 | 8.2 | 4.6 | 0 | 0 | 0 | 0 | 0 |

Table 1 Preferred anisotropic velocity models based on >400 test models, suggesting that the 20-km-thick lower crust is highly anisotropic ($\delta Vs > 10\%$) in the Kaapvaal and Zimbabwe cratons, and moderately anisotropic (7% throughout the crust) in the Limpopo Belt. The difference in plunge angle explains variation in relative arrival time between radial SV and transverse SH phases: SH is faster than SV in Limpopo Belt and SV is the fastest in the cratons.

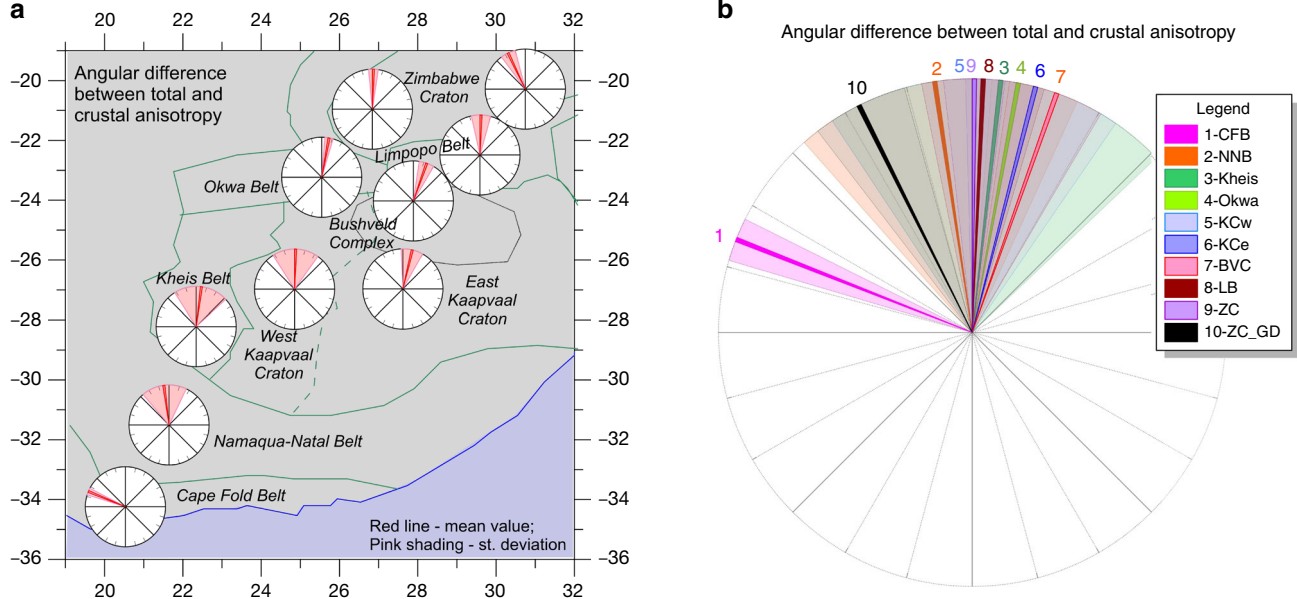

**Fig. 5** Difference between fast axis directions for SKS and RF-determined anisotropy. **a** Divided into ten tectonic units. Red lines mark average difference, and pink shading marks standard deviation in the rose diagrams, where zero difference corresponds to the north direction. **b** All differences plotted in the same rose diagram. The SKS- and RF directions are generally parallel, except for the Cape Fold Belt and to some degree the northern part of the Zimbabwe Craton around the Great Dyke. *Abbreviations*: CFB—Cape Fold Belt, NNB—Namaqua–Natal Belt, KB—Kheis Belt, OB—Okwa Belt, KCw—West Kaapvaal Craton, KCe—East Kaapvaal Craton, BVC—Bushveld Complex, LB—Limpopo Belt, ZC—Zimbabwe Craton, ZC_GD—Zimbabwe Craton around the Great Dyke.

belts (Fig. 1, Supplementary Fig. 7), which indicates a generic connection. We notice that the swarms consist of a large number of individual thin dykes that may generate a structural type of anisotropy (SPO) with fast axis parallel to the dyke direction[42]. Unlike mantle dykes, the seismic velocity of mafic dykes in the upper crust is usually much higher than in the surrounding host rock of granitic origin[43]. Possible presence of several dyke generations in each swarm and incomplete sampling for isotope dating[44] precludes precise dating of the dykes. However, a striking contrast between the orientation of the ~180 Ma Okavango dyke swarm[45] and the fast axis in the Zimbabwe Craton (Fig. 1d) suggests that the anisotropy is pre-Jurassic in this region.

We speculate that pre-existing lithosphere weakness fabric both determine the fast anisotropy direction and also may have guided the direction of the dykes. The overall weakness fabric determines anisotropy to large depths. The dykes contribute to the total anisotropy with fast axis parallel to the horizontal component of the original fast axes, thereby contributing constructively to the crustal anisotropy amplitude. The initial anisotropy in the host rock is maintained with a dip component of the fast direction. Therefore, our modelled dip of the fast axis may be more horizontal than the initial direction of the fast axis due to the effect of the later dyke swarms.

Mica schist, tonalitic gneiss and amphibolite are the main candidates for producing strong LPO anisotropy in the middle-to-lower crust[42,46]. Geochronology indicates that the cratonic crust has experienced a dynamic metamorphic history since its formation around 3.2 Ga[5,47]. The lower crust may consist of a granite gneiss domain composed of massive-to-foliated metamorphosed granitoid above a gneiss complex with an intrusive, transitional contact zone[4,48]. Biotitic gneisses display significant

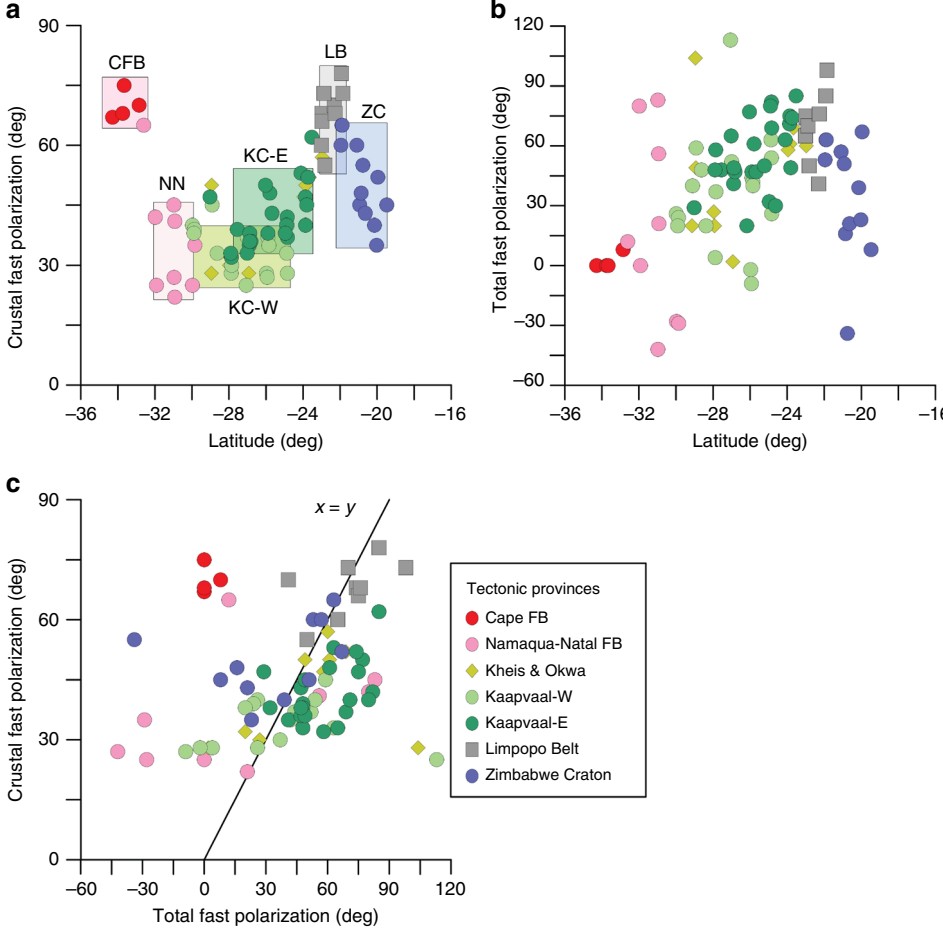

**Fig. 6** Observed fast polarisation directions ($\varphi$) versus latitude for **a** crustal anisotropy (our results), and **b** total anisotropy (from SKS splitting[21]); **c** cross-plot of $\varphi$ for crustal anisotropy from RF and for SKS analysis. Colours indicate tectonic province. Abbreviations as in Fig. 5.

anisotropy ($\Delta V$ ~0.3 km/s) with polarisation planes related to the tectonic fabric[49]. We envisage that past deformation modified the lower crust by aligning the fast axes of the gneiss texture with steeply inclined features.

Independent studies indicate craton-wide events between 3.1 and 2.7 Ga involving magmatism, thrusting and metamorphism that may imply structural and thermal remobilisation of the lithosphere[50]. These episodes may be key to the creation of the observed lithospheric anisotropy. Transient thermal pulses are observed in cratonic sapphirine granulite xenoliths from Kaapvaal dated[5] to 2.723 Ga, which may be associated with the Ventersdorp flood basalt and lithospheric thinning[47].

Our analysis demonstrates the presence of unexpected, strong crustal anisotropy and indicates that the anisotropy in the depleted Archaean cratonic mantle[21,51] may be up to 30–55% weaker than that previously believed. Determination of the relative strength of the anisotropy between crust and mantle from comparison of SKS- and RF-splitting parameters depends on the relative difference in directions of the fast axes in the two layers. The effect of the strong crustal anisotropy is both to rotate the observed fast axis in SKS-splitting studies towards the crustal direction and to weaken the strength of the total anisotropy. Therefore, the total mantle anisotropy may be relatively larger than that indicated by this upper limit if the orientations deviate substantially. However, the close agreement in fast axes orientation determined from RF and SKS splitting indicates that the difference between fast axes in the crust and mantle lithosphere is small. The overall similarity in fast polarisation direction between

the anisotropy determined from SKS splitting[21] and the crustal anisotropy of the cratons (this study), therefore, indicates that the crust and mantle have been coupled since the anisotropy formed. This indicates stability of the lithosphere since the time of the last tectonomagmatic events, and provides independent support to xenolith studies[5] for long-term stability of the upper 150 km of the lithosphere in the Kalahari Craton.

Previous studies based on the same seismic data did not recognise or model the crustal anisotropy[21–23], assuming that it is a negligible component of the total anisotropy. We propose that much of the crustal anisotropy was acquired during craton formation and evolution involving collisional tectonics, magmatic activity and thermal recycling. These reworking processes changed the chemistry of the lower crust to increase the anisotropy and produce metamorphosed bands (mica-foliated gneiss) with intermediate (granitoid gneiss) composition.

The present plate motion direction is parallel to the fast axes determined for the crust by our RF study and SKS- splitting results[21] for a large part of the Kalahari Craton, but they significantly deviate in the Limpopo Belt and especially in the Cape Fold Belt. Inversion of the SASE data for the depth variation of S-wave anisotropy in the Limpopo Belt indicates that the fast direction changes from ca. 90° to ca. 40° at around 160-km depth[23]. The deep fast direction is close to the present plate motion direction, and the authors infer that this coincidence is caused by present deformation in the sublithospheric mantle, whereas the shallow direction corresponds to frozen-in anisotropy in the mantle lithosphere[23].

The SKS-splitting results[21] indicate very weak anisotropy in the Cape Fold Belt and the Namaqua–Natal Fold Belt, whereas our analyses show strong anisotropy in the crust (Fig. 1). One may speculate that the weak SKS anisotropy may be due to a highly heterogeneous depth distribution of the fast direction in the mantle, which would tend to reduce the total SKS-splitting anisotropy[40]. It also indicates that plate motion parallel anisotropy in the asthenosphere may not be dominant in southern Africa because, otherwise, strong SKS splitting would be observed also in these fold belts.

However, the parallel fast axes for the crust and from SKS-splitting analyses indicate that the major part of the measured SKS splitting originates in the lithosphere, whereas the contribution from the asthenosphere is smaller. This interpretation is also supported by the observation that the SKS-splitting directions[21] deviate substantially from the plate motion direction in the Limpopo Belt and the Cape Fold Belt. The overall coincidence in fast axes between the crustal and the SKS-splitting anisotropy within each tectonic block indicates long-term coupling between crust and mantle lithosphere, with the indication that crust and mantle lithosphere have remained one solid entity since cratonisation. This finding may have fundamental implications for the degree to which mantle flow can affect lithosphere deformation, for the rheological structure at the lithosphere–asthenosphere transition, and for the strength of the lithospheric crust and mantle.

## Methods

**Receiver functions**. We apply the RF method in the LQT version[31,32], which is based on classic ray theory and suppresses the almost vertical P-wave motion energy by decomposing the seismic wavefield into the L, Q and T components. Conversion at boundaries between different layers for both incident P- and S waves is described by the Zoeppritz equations[52]. P-wave RFs are calculated by isolating the P-to-S conversions[31,53] by carrying out a deconvolution procedure of the longitudinal components from the horizontal components. Deconvolution may be done in the time or the frequency domain[54,55]. We assume incident plane teleseismic waves and we calculate the resulting waveform as a band-limited impulse response at the stations as a function of slowness.

It is possible to estimate the depth to discontinuities and the average velocity above it, if the data cover an adequate epicentral distance interval[56,57] and by use of reverberations and conversions[58]. The travel-time difference between the Pds phase (the P- to S-converted wave) and the direct P wave depends on the depth and dip of the converting discontinuity, the P- and S-wave velocity structure between the discontinuity and the seismic station and the epicentral distance between station and source.

The LQT method[31] in the version by Yuan et al.[32] is based on rotation of the geographically oriented seismograms into ray coordinates defined by the L, Q and T axes. This decomposes the wavefield into the L-component that is the subvertical wave (P), the Q-component that is the radial vertically polarised shear wave (SV) and the T-component that is the transverse horizontally polarised shear wave (SH). It is a common assumption[32] that the P and SV components are orthogonal: they are determined by rotating the vertical and radial waveforms to obtain a minimum P-component amplitude at the mean converted S arrival time. We use all radial and transverse-component RFs in the analysis[59]. Anisotropic velocity structure leads to different velocity for the two S phases that therefore will have different arrival times and their waveforms will further show back-azimuthal variation, including phase reversals.

Straightforward frequency-domain deconvolution is often unstable due to spectral holes and requires stabilisation by either pre-whitening[60] or water-level[53] algorithms. We choose, alternatively, to apply a stable procedure based on iterative, time-domain spiking deconvolution[59,61] with pre-whitening to stabilise the filtering to isolate the phases. The Pds phases are further enhanced by stacking the deconvolved signals using the appropriate moveout corrections for different slowness[32].

Iterative time-domain deconvolution is a stable procedure even for complex signals. However, the response at the receiver will always depend on the complexity of the structure in the medium. Simple structures generally lead to better RF images[61]. In principle, the model incidence angles could be calculated directly from, e.g., the standard IASP91 velocity structure, but we choose to determine the real rotation angles by an iterative procedure where we minimise the P component on the S components. We further apply band-pass filtering in various frequency intervals to resolve different parts of the crustal structure[29].

**Synthetic RFs**. We document for the first time that very strong anisotropy may reside in the crust of cratonic regions by demonstrating that radial $P_{SV}$ and transverse $P_{SH}$ RFs have distinct different arrival times. By calculating synthetic

RFs for anisotropic models, we further determine the strength and the plunge angle for the fast axes within layers in the Earth. For this we apply the method derived by Frederiksen and Bostock[35]. This method calculates synthetic RFs as functions of back azimuth for models that include anisotropy. The models are defined by layer thicknesses, velocities and anisotropy fast axes and strengths. We plot the synthetic and observed RFs in sections versus back azimuth, which enables us to make a visual comparison between synthetic and observed sections. Quantitative comparison is unfortunately not feasible due to the relative high noise level in the observed RFs versus back azimuth, as has for long been accepted in the seismological community for active tectonic regions[25–28,62,63]. In Supplementary Figs. 1–6 we provide data from a systematic search of anisotropic models that may explain our observations. The search demonstrates a high degree of nonlinearity between models and observations, which show that our derived models are non-unique, although representative.

## Data availability

All seismic data used in this study are openly available at the IRIS web page http://ds.iris.edu/ds/nodes/dmc/data/.

## Code availability

All seismic codes used in this study are openly available at the IRIS web page http://ds.iris.edu/ds/nodes/dmc/data/.

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

## Acknowledgements

We thank the IRIS/PASSCAL Data Management Centre for providing the SASE seismic data and the seismic codes used in this study. Discussions with Lev Vinnik improved the paper. This study was supported by grants FNU10-083081 to I.M.A. and FNU11-104254 to H.T. from the Danish Research Council. Partially supported by the MOST special funds for GPMR State Key Laboratory (GPMR2019010).

## Author contributions

H.T. developed the concept and methodology of the study. M.Y. assembled the data set and made the calculations of receiver functions and the supporting resolution tests in discussion with H.T. I.M.A. developed the interpretation. All authors intensively discussed the results and jointly developed implications. All authors contributed to writing the paper and making the illustrations.

## Competing interests

The authors declare no competing interests.
