## [Peer Review File · Nature Communications]

Reviewers' comments:

Reviewer #1 (Remarks to the Author):

The paper presents convincing evidence for the existence of azimuthal anisotropy in the crust of the Kalahari Craton. It also demonstrates that the fast-propagation directions are consistent within tectonic blocks and are similar to the published fast-propagation directions from SKS splitting analysis. The observations are related to the important outstanding problems of the formation and evolution of cratons.

The new measurements present a significant addition to the existing anisotropy observations. The well-known SKS splitting measurements in this region have been used to draw very different conclusions, as noted in the paper. The SKS measurements are cumulative measures of anisotropy in the asthenosphere, mantle lithosphere and the crust, but different authors chose to emphasize the contributions of the asthenosphere only or, alternatively, of the mantle lithosphere only. Surface waves can resolve the depth dependence but at the expense of the horizontal resolution. The new receiver-function measurements presented here average over the thickness of the crust only and provide new constraints on the depth distribution of anisotropy. And their high horizontal resolution allows the authors to demonstrate the coherence of the anisotropy within tectonic blocks.

The evidence for anisotropy is convincing and important, and I will be looking forward to seeing it published. My comments and suggestions are mainly on the interpretation of the measurements.

1. The similarity of the fast propagation directions in the crust only (from RF) and in the crust and upper mantle in total (from SKS) can be shown more convincingly by an average angular misfit between the measurements of the two types over all the stations and by a histogram of the angular misfits from different stations.

2. The authors list the likely contributions to the crust-average anisotropy measured, including the alignment of anisotropic minerals in the lower crust, aligned cracks in the upper crust, and aligned dykes. The orientation and strength of the anisotropy created by each of these elements will have an effect on the crust-average anisotropy. I would suggest to estimate how the average crustal fast-propagation directions (Figure 1) depend on and reflect these different contributions to the total.

3. The SKS splitting integrates anisotropy in the entire crust and upper mantle, as the authors point out. If the contribution of the crust to the splitting is significant, as the authors also point out, rightly, then to what extent could the crustal anisotropy determine the distribution of SKS splitting fast azimuths? Because the authors base their conclusions on the similarity of the crustal and SKS anisotropy, this is important to estimate.

4. The paper concludes with a suggestion that the present plate motion does not create anisotropy in the sub-lithospheric mantle beneath southern Africa, with a hint for broader implications from this. I could not see how this was supported by the new measurements or by arguments in the paper.

Reviewer #2 (Remarks to the Author):

This article presents new results on seismic anisotropy within the lower crust of the Kaapvaal craton based on receiver function analysis. Based on the comparison between these data and previous SKS splitting data obtained in the same stations, they propose that a large part of the

observed seismic SKS splitting stems from the lower crust and that crustal and mantle anisotropy orientations are coherent, implying a coupled evolution for more than 2 Ga.

The idea is interesting, but the presentation of the actual receiver function observations and of the method used to infer seismic anisotropy fast polarization directions and delay times accumulated in the lower crust based on these data is not clear. By consequence, it is difficult to assess the robustness of the deductions of seismic anisotropy in the lower crust.

Measuring seismic anisotropy based on receiver function analysis is not as straightforward as SKS splitting. Therefore, the authors should present in detail the method, the assumptions behind it, and the limitations. One point in the presentation of the seismic anisotropy results in the present ms. is particularly disturbing: The use of the denominations SH and SV are fine for describing the different polarizations of horizontally travelling waves, like surface waves. However, the converted S-waves analyzed in this study do not travel horizontally and, hence, SH and SV have no clear meaning. The split quasi-shear waves should be described in terms of radial (the wave polarized in the backazimuth direction) and transverse components, like SKS.

It is also important to note that PS receiver function anisotropy analyses do not sample exactly the same directions than SKS splitting (vertical). Seismic anisotropy of rocks is a 3D property. By consequence, Moho converted PS and SKS waves "see" different elastic properties. Depending on the rock type and on the orientation of the deformation tensor that produced the elastic anisotropy, the difference may be important. The difference in propagation paths for the two types of waves should therefore be explicitly taken into account if one wants to compare the two types of data. However, this point seems to have been neglected in the present study.

By consequence, although the results seem interesting, I cannot recommend the publication of the article. Below I list a series of other points that should also be considered in revising the article.

1- A more detailed presentation of the seismic data used is necessary:

- Please show a figure with the azimuthal and distance distribution of the data and PS conversion points.
- Incidence angle for the converted waves depends on the distance, this means that stacking data from different distances may result in mixing data that sampled different directions in the rock structure.
- In the receiver function figures, please identify the different converted phases and multiples, so that the non-specialist reader may easily access what is the significant data.
- The observations and the method that allow defining the dip of the structure producing the anisotropy should be clearly presented and discussed. Effect of a possible inclination or topography of the discontinuities (dipping Moho or Moho steps) on the RF data should be discussed and the arguments for discriminating such effects from a dipping anisotropy presented.

2- The method used by the authors to choose their preferred model of fast direction and delay time in the lower crust is not clear... Visual comparison is a very qualitative way of evaluating the fit of a model to data. When I analyze visually the figures, I see as much difference as similarity between the data and the different synthetics. Is there a more quantitative way to evaluate the quality of the fit obtained using different synthetic models?

3- Intracrustal interface: There is a clear difference in signal between waves converted at Moho and the top of the lower crust. The second signal is much weaker and less anisotropic and it displays changes in polarity at different backazimuths. I would interpret this as indicating a variation of the anisotropy with depth, with the strongest signal coming from the lower crust... This point should be discussed.

4- Interpretation of the role of the dykes on the observed seismic anisotropy is not clear. Why would a fully crystallized dyke produce seismic anisotropy? Contrast in seismic properties of a

crystallized basalt relative to the host-rock is usually too weak.

5- The interpretation that the anisotropy of the cratonic mantle is the difference between the SKS and receiver function determined anisotropy is not unique. Another possible interpretation is that the anisotropy directions in the mantle differ from the crustal ones and that the SKS splitting data is the composite record of a depth-variable anisotropy. In this case, the mantle contributions may be higher than the one explaining the arithmetic difference between the SKS and Moho receiver function delay times.

6- The discussion of the tectonic evolution of the Kaapvaal lithosphere is not clear. On one side in the introduction the authors propose that the entire cratonic lithosphere (crust and mantle) was stabilized at 3 Ga, and on the other, in the discussion: they propose "craton-wide events between 3.1 and 2.7 Ga involving magmatism, thrusting, and metamorphism that may imply structural and thermal remobilization of the lithosphere". The use of Ventersdorp data is also surprising – this is an impact structure.

7- Other "geodynamical" statements are proposed without been justified by a solid reasoning. For instance:

a- Which are the implications of long-term crust mantle coupling for the rheology of the lower lithosphere and (for) the effect of mantle flow on the lithosphere deformation? This is proposed at the end of the abstract but not really discussed in the article.

b- Parallelism of the seismic anisotropy directions and dyke swarms: If it is real, what does it mean in terms of physical processes?

8- The comparison between the inferred anisotropy in the lower crust and outcropping crustal structures should be more throughout – distinguishing the different types and episodes of deformation among the outcropping structures.

9- Reference 4 does not describes xenolith data.

Reviewer #3 (Remarks to the Author):

This paper provides evidence that crust of the cratonic lithosphere has intense deformation that has close correlation with deformation of the mantle part. And such deformation occurred during the time of the lithosphere formation.

This is a new concept and is a paradigm change: in the past, contribution from crust to seismic anisotropy was considered to be negligible.

This discovery has a few impacts to some of the fundamental issues of cratonic lithosphere formation and the nature of deformation in the continental lithosphere.

If crust and mantle deformed in a similar way, it would suggest deformation in the vertical shear rather than the horizontal shear. If this is true, then our interpretation of anisotropy of the mantle part of the continental lithosphere needs to be modified (in most of interpretation of seismic anisotropy, horizontal shear is assumed).

I consider that this is an important contribution to the community to provide a new set of observations that force us to modify our view of deformation of the cratonic lithosphere.

However, I have some concern when the authors say that the long-term stability of the continental lithosphere depends on the nature of crust-mantle coupling. My understanding is that the stability of the cratonic mantle is mainly controlled by the nature of deformation of the main part of the

lithosphere, i.e., the mantle that has not much related to the nature of crust-mantle coupling. Statement on this point needs to be "softened".

Reviewer #1 (Remarks to the Author):

The paper presents convincing evidence for the existence of azimuthal anisotropy in the crust of the Kalahari Craton. It also demonstrates that the fast-propagation directions are consistent within tectonic blocks and are similar to the published fast-propagation directions from SKS splitting analysis. The observations are related to the important outstanding problems of the formation and evolution of cratons.

The new measurements present a significant addition to the existing anisotropy observations. The well-known SKS splitting measurements in this region have been used to draw very different conclusions, as noted in the paper. The SKS measurements are cumulative measures of anisotropy in the asthenosphere, mantle lithosphere and the crust, but different authors chose to emphasize the contributions of the asthenosphere only or, alternatively, of the mantle lithosphere only. Surface waves can resolve the depth dependence but at the expense of the horizontal resolution. The new receiver-function measurements presented here average over the thickness of the crust only and provide new constraints on the depth distribution of anisotropy. And their high horizontal resolution allows the authors to demonstrate the coherence of the anisotropy within tectonic blocks.

The evidence for anisotropy is convincing and important, and I will be looking forward to seeing it published. My comments and suggestions are mainly on the interpretation of the measurements. Thank you for this constructive review. We agree and emphasise that this paper presents for the first time significant crustal anisotropy in cratonic crust.

1. The similarity of the fast propagation directions in the crust only (from RF) and in the crust and upper mantle in total (from SKS) can be shown more convincingly by an average angular misfit between the measurements of the two types over all the stations and by a histogram of the angular misfits from different stations.

Done. Thank you for this good suggestion. New figures 5ab and S7ab.

2. The authors list the likely contributions to the crust-average anisotropy measured, including the alignment of anisotropic minerals in the lower crust, aligned cracks in the upper crust, and aligned dykes. The orientation and strength of the anisotropy created by each of these elements will have an effect on the crust-average anisotropy. I would suggest to estimate how the average crustal fast-propagation directions (Figure 1) depend on and reflect these different contributions to the total.

We have added to the discussion of these aspects, although we find that it is impossible to provide a quantitative calculation of the relative contributions. However, if the dykes are features of the upper crust they can only contribute to the measured anisotropy above the mid-crustal converter. Their strike is parallel to the overall crustal anisotropy in most areas, except for the area with the very special and huge Great Dyke, which indicate that the crustal fabric and mineralogy may determine both the main part of the anisotropy and, perpendicularly, the direction of minimum strength.

3. The SKS splitting integrates anisotropy in the entire crust and upper mantle, as the authors point out. If the contribution of the crust to the splitting is significant, as the authors also point out, rightly, then to what extent could the crustal anisotropy determine the distribution of SKS splitting fast azimuths? Because the authors base their conclusions on the similarity of the crustal and SKS anisotropy, this is important to estimate.

The most authoritative study of the effect of variable anisotropy with depth by Saltzer et al. (2000) shows that the orientation of SKS observed anisotropy tends to be weighted towards the orientations in the upper parts of the model for long wavelength variations. Very short-wavelength

variation in anisotropy direction may even make measurement of anisotropy by SKS splitting impossible by reducing the amplitudes to below detection limit. However, the net effect is that the SKS fast axis rotates from the mantle fast direction toward the fast axis of the crustal anisotropy. We observe alignment which indicate that the crustal and mantle directions cannot differ substantially. We have added to the discussion of this point.

4. The paper concludes with a suggestion that the present plate motion does not create anisotropy in the sub-lithospheric mantle beneath southern Africa, with a hint for broader implications from this. I could not see how this was supported by the new measurements or by arguments in the paper.

We did express ourselves unclear on this point. We have therefore rephrased the discussion. The meaning is that that the present plate motion apparently is parallel to the observed anisotropy in the crust and from SKS splitting in most parts of the Kalahari craton, and this may happen by chance. The plate motion and fast axis directions differ in the Limpopo belt, so the alignment in the asthenosphere cannot be the primary cause of mantle anisotropy. Further, the SKS anisotropy amplitude is extremely small in the Cape and Namaqua-Natal fold belts, where we, nevertheless, observe strong crustal anisotropy by the RFs. This indicates that the mantle anisotropy is highly variable, if it exists, and that the asthenospheric anisotropy may not be dominant, because otherwise the situation should be as in the craton part. This is the reason that we think that asthenosphere anisotropy is not the dominant part.

Reviewer #2 (Remarks to the Author):

This article presents new results on seismic anisotropy within the lower crust of the Kaapvaal craton based on receiver function analysis. Based on the comparison between these data and previous SKS splitting data obtained in the same stations, they propose that a large part of the observed seismic SKS splitting stems from the lower crust and that crustal and mantle anisotropy orientations are coherent, implying a coupled evolution for more than 2 Ga.

The idea is interesting, but the presentation of the actual receiver function observations and of the method used to infer seismic anisotropy fast polarization directions and delay times accumulated in the lower crust based on these data is not clear. By consequence, it is difficult to assess the robustness of the deductions of seismic anisotropy in the lower crust.

Measuring seismic anisotropy based on receiver function analysis is not as straightforward as SKS splitting. Therefore, the authors should present in detail the method, the assumptions behind it, and the limitations. One point in the presentation of the seismic anisotropy results in the present ms. is particularly disturbing: The use of the denominations SH and SV are fine for describing the different polarizations of horizontally travelling waves, like surface waves. However, the converted S-waves analyzed in this study do not travel horizontally and, hence, SH and SV have no clear meaning. The split quasi-shear waves should be described in terms of radial (the wave polarized in the backazimuth direction) and transverse components, like SKS.

We have added a Method Section. We use SV and SH in order to be consistent with the nomenclature used by Frederiksen and Bostock (2000) in the paper describing the modelling software used here. It is customary to use the terms SV and SH in papers on anisotropy determined from receiver functions. We have added an explanatory sentence to avoid misunderstandings.

It is also important to note that PS receiver function anisotropy analyses do not sample exactly the same directions than SKS splitting (vertical). Seismic anisotropy of rocks is a 3D property. By consequence, Moho converted PS and SKS waves "see" different elastic properties. Depending on the rock type and on the orientation of the deformation tensor that produced the elastic anisotropy, the difference may be important. The difference in propagation paths for the two types of waves should therefore be explicitly taken into account if one wants to compare the two types of data. However, this point seems to have been neglected in the present study.

We respectfully disagree with the reviewer. A quick look at a global traveltimes plot would have shown that ray parameters for SKS and Ps are similar, which means that the two waves travel in the same direction through the crust, with an angle of around 10° from the vertical. Therefore, they do sample the same part of the crust at each station, and they do not see the properties differently. We had not neglected this point, but found it obvious. Vinnik et al. (2012) imply the same consideration when carrying out joint inversion of SKS and receiver functions.

By consequence, although the results seem interesting, I cannot recommend the publication of the article. Below I list a series of other points that should also be considered in revising the article.

We suppose that we have argued convincingly to the above critics, which therefore appears irrelevant to our study.

1- A more detailed presentation of the seismic data used is necessary:

- Please show a figure with the azimuthal and distance distribution of the data and PS conversion points.

Added

- Incidence angle for the converted waves depends on the distance, this means that stacking data from different distances may result in mixing data that sampled different directions in the rock structure.

The variation in incidence angle in the crust is between 8 and 14° which is well within the uncertainty that we estimate. The related difference in distance between piercing points at the Moho is 4 km which is well within the size of the Fresnel Zone, so the sampling is well constrained within the uncertainties estimated. We now include a figure showing the distribution of sources, which show that the vast majority of the sources occurred at epicentral distances between 70° and 90°, which correspond to incidence angles between 8 and 10° and Moho sampling roughly within 1 km. Therefore, there is very little mixing and the effects are much smaller than the estimated uncertainties. We believe that those readers who will pay attention to these details will already be aware of them. We thank the reviewer for requesting the new figure with epicentral distribution of the sources.

- In the receiver function figures, please identify the different converted phases and multiples, so that the non-specialist reader may easily access what is the significant data.

Done

- The observations and the method that allow defining the dip of the structure producing the anisotropy should be clearly presented and discussed. Effect of a possible inclination or topography of the discontinuities (dipping Moho or Moho steps) on the RF data should be discussed and the arguments for discriminating such effects from a dipping anisotropy presented.

We refer to our paper (Youssof et al. 2013) where we present Moho depths in the study area, from which we have calculated the gradient of the Moho topography. The result shows that the major part of the Kalahari craton has very small gradient with exceptions at around 20 stations at Bushveld, the Great Dyke area, and eastern Limpopo Belt, as well as in the Cape Fold Belt. As there is no significant slope on Moho at the majority of stations, we can conclude that in general we are modelling true anisotropy and not structural effects.

2- The method used by the authors to choose their preferred model of fast direction and delay time in the lower crust is not clear... Visual comparison is a very qualitative way of evaluating the fit of a model to data. When I analyze visually the figures, I see as much difference as similarity between the data and the different synthetics. Is there a more quantitative way to evaluate the quality of the fit obtained using different synthetic models?

To our knowledge, no study of anisotropy by use of receiver functions has applied quantitative measures of misfit between synthetic and observed sections. It is not possible to do, considering the signal to noise level. We have now annotated the sections. We refer to the many existing papers on crustal anisotropy in e.g Tibet, which all rely on similar comparison.

3- Intracrustal interface: There is a clear difference in signal between waves converted at Moho and the top of the lower crust. The second signal is much weaker and less anisotropic and it displays changes in polarity at different backazimuths. I would interpret this as indicating a variation of the

anisotropy with depth, with the strongest signal coming from the lower crust... This point should be discussed.

The reviewer's observation is right, it is agreement with the original text of our manuscript and it is part of the preferred model (Table S1, now Table 1). We do discuss that the anisotropy is strongest in the lower crust, and that this observation may have implications for the understanding of the importance of the other observation that the fast axes are aligned with the strike of dyke swarms.

4- Interpretation of the role of the dykes on the observed seismic anisotropy is not clear. Why would a fully crystallized dyke produce seismic anisotropy? Contrast in seismic properties of a crystallized basalt relative to the host-rock is usually too weak.

We have added to this discussion, as it may represent a hen-egg problem. However, the reviewer is not right about the velocity contrast between mafic dykes and crustal host rocks. This contrast is very large between ca. 6 km/s and 6.7-7.0 km/s, actually the largest velocity contrast observed in the crystalline crust. The reviewer may have their experience from studying basaltic intrusions in the mantle where it is correct that the contrast can be negligible.

5- The interpretation that the anisotropy of the cratonic mantle is the difference between the SKS and receiver function determined anisotropy is not unique. Another possible interpretation is that the anisotropy directions in the mantle differ from the crustal ones and that the SKS splitting data is the composite record of a depth-variable anisotropy. In this case, the mantle contributions may be higher than the one explaining the arithmetic difference between the SKS and Moho receiver function delay times.

It is correct that the mantle anisotropy may be larger than the arithmetic difference between crustal and SKS strength, mainly if the fast axes directions are substantially different. However, in this case the SKS and RF directions will differ, even taking into account that the crustal anisotropy tends to rotate the SKS fast direction into the crustal fast direction. Therefore, we find that the similar directions indicate that the mantle and crustal anisotropy have (sub-) parallel fast axes.

6- The discussion of the tectonic evolution of the Kaapvaal lithosphere is not clear. On one side in the introduction the authors propose that the entire cratonic lithosphere (crust and mantle) was stabilized at 3 Ga, and on the other, in the discussion: they propose "craton-wide events between 3.1 and 2.7 Ga involving magmatism, thrusting, and metamorphism that may imply structural and thermal remobilization of the lithosphere". The use of Ventersdorp data is also surprising – this is an impact structure.

We actually write that the craton formed and stabilised during the Archaean which means until 2.5 Ga– not at 3 Ga as the reviewer states and which we did not write. We do describe the mentioned events that took place around the amalgamation and further evolution of the Kalahari craton, and there is no discrepancy in this.

We suspect that the reviewer confuses the Ventersdorp flood basalts (ca. 2.7 Ga magmatic event) with the Vredefort crater (ca. 2.0 Ga impact structure), which brought Ventersdorp magmatic rock to the surface. Our description is also here in accordance with the literature, unlike the review.

7- Other "geodynamical" statements are proposed without been justified by a solid reasoning. For instance:

a- Which are the implications of long-term crust mantle coupling for the rheology of the lower

lithosphere and (for) the effect of mantle flow on the lithosphere deformation? This is proposed at the end of the abstract but not really discussed in the article.

b- Parallelism of the seismic anisotropy directions and dyke swarms: If it is real, what does it mean in terms of physical processes?

We have added to the discussion of these aspects, following comments from reviewer 1.

8- The comparison between the inferred anisotropy in the lower crust and outcropping crustal structures should be more throughout – distinguishing the different types and episodes of deformation among the outcropping structures.

We are uncertain about to which outcropping crustal structures the reviewer refers. We do not discuss outcrops. We do discuss our observations in relation to generally accepted divisions of the crustal age areas in southern Africa. We may refer the reviewer to our paper by Youssof et al. (2013), referenced in this paper, where we provide a detailed discussion of the relation between crustal structure and crustal provinces.

9- Reference 4 does not describes xenolith data.

We have improved the wording to include surface mapping and chronology; however, the reference actually uses results from xenolith data in the analysis.

Reviewer #3 (Remarks to the Author):

This paper provides evidence that crust of the cratonic lithosphere has intense deformation that has close correlation with deformation of the mantle part. And such deformation occurred during the time of the lithosphere formation.

This is a new concept and is a paradigm change: in the past, contribution from crust to seismic anisotropy was considered to be negligible.

This discovery has a few impacts to some of the fundamental issues of cratonic lithosphere formation and the nature of deformation in the continental lithosphere.

If crust and mantle deformed in a similar way, it would suggest deformation in the vertical shear rather than the horizontal shear. If this is true, then our interpretation of anisotropy of the mantle part of the continental lithosphere needs to be modified (in most of interpretation of seismic anisotropy, horizontal shear is assumed).

I consider that this is an important contribution to the community to provide a new set of observations that force us to modify our view of deformation of the cratonic lithosphere.

We thank the reviewer for this presentation of some of the wide ranging perspectives of our finding.

However, I have some concern when the authors say that the long-term stability of the continental lithosphere depends on the nature of crust-mantle coupling. My understanding is that the stability of the cratonic mantle is mainly controlled by the nature of deformation of the main part of the lithosphere, i.e., the mantle that has not much related to the nature of crust-mantle coupling. Statement on this point needs to be "softened".

We agree with the reviewer that the long term stability of the cratonic lithosphere primarily depends on the rheology of the mantle lithosphere, as this is the strong part. Our point about the coupling between the mantle and crustal lithosphere is, that this is necessary to keep the crust and mantle lithosphere together as one unit over geologic time. We have updated the text where this aspect is discussed and softened the text in the abstract, such that it should now be clear that the rheology of the mantle is central and that the coupling between crust and mantle lithosphere also has importance.

Reviewers' comments:

Reviewer #2 (Remarks to the Author):

The authors have answered satisfactorily to most comments by myself and the other reviewer. I realize by reading their answers that some of my concerns were not justified. I recommend therefore the publication.

Nevertheless, I really think the authors should use the terms radial and transverse for describing the anisotropic converted waves instead of SV and SH, since the latter terms have no actual physical meaning for a vertically propagating wave. Their use is therefore confusing for non-specialists in RF. However, if they insist in using SV & SH for coherence with previous RF studies, as argued in the response to the reviewers, they should complete the statement added to the revised version by clearly stating that the studied waves have almost vertical propagation directions and, by consequence, SV and SH have not a real significance.

Below are some minor comments that I hope will be useful in preparing the final version:

1- The verb is missing in the 1st phrase of the last paragraph of the results section. "which indicate the directions of the fast polarization axes (ϕ) in the lower and upper crust" ??? I guess that the missing verb is "differ"?

2- Dykes are roughly vertical planar structures. If the basalts have a marked contrast in seismic velocity with the surrounding crustal rocks, a dyke swarm will produce anisotropy with a hexagonal symmetry with a slow axis normal to the dykes' trend. However a dyke swarm cannot produce the highly inclined symmetry axes of anisotropy proposed for most of the crust based on the RF data (6-40 km depth, cf. Table 1).

3- Similarity in the fast polarization axes in the crust and the lithospheric mantle. I agree that if the two directions were markedly different between the crust and the mantle, the apparent SKS splitting should deviate from the fast direction in the crust. However, the strongest evidence for consistency of the fast directions along the SKS path would be absence of back-azimuthal variation in the observed fast polarization directions and delay times for the SKS. From what I remember the back-azimuthal coverage is not very good, but maybe for some stations this point can be tested?

Reviewer #4 (Remarks to the Author):

As I was contacted by the editor to replace one of the reviewer for the second round, I will try not to cover the topics that have been already raised and I will not provide a very detailed review.

This paper addresses a difficult topic, the separation of crustal and mantle contribution to seismic anisotropy. Indeed the vertical resolution of body waves is always rather poor. The authors try to circumvent the problems by using Ps crustal conversions to isolate the crustal contribution. But as correctly pointed by one of the reviewer, it is sometimes difficult to separate the effect of anisotropy from those coming from dipping isotropic structures. The main issue is that it is also unclear what causes crustal anisotropy, in contrast to mantle anisotropy which presumably mainly comes from lattice-preferred orientation of olivine aggregates. Therefore the apparent fast direction in the crust may not necessarily be the principal shear direction as in the mantle. This caveat could perhaps be discussed.

In addition to properly model the effects of seismic anisotropy on seismic waveforms one needs to consider the complete elasticity tensor. Indeed, as pointed out by the reviewers shear wave splitting and Ps conversions are not sensitive to the same elasticity coefficients so this has to be kept in mind when interpreting the results of modeling. Perhaps my main concern is that in any

case the synthetic seismograms bear only a vague resemblance to the observed data. I think that quantifying the waveform misfit might be important to discriminate between the different models to try to convince the reader that indeed these models indeed fit the data and are robust.

Regarding the interpretation, the apparent coherence between crustal and mantle anisotropy could indeed suggest some vertically coherent deformation of the cratonic lithosphere of the Kaapvaal craton. Since the overall anisotropy is weak, this would suggest a very strong and hard to deform cratonic lithosphere, which raises the question as the rheology of this lithosphere at Archean times with a much higher geotherm.

Finally, I think that the bibliography is uncomplete. I would like to see some more thorough comparisons with the results of surface wave azimuthal anisotropy studies in that region. There is also a nice paper by Fouch et al. (2004) which evidenced large small-scale variations of apparent splitting in the Kaapvaal craton. The results of this study demonstrate that there are also important lateral variations of seismic anisotropy in that region, and that stratification is not the complete story.

Reviewer #2 (Remarks to the Author):

The authors have answered satisfactorily to most comments by myself and the other reviewer. I realize by reading their answers that some of my concerns were not justified. I recommend therefore the publication.

Thank you for critically assessing our arguments and for the new constructive suggestions.

Nevertheless, I really think the authors should use the terms radial and transverse for describing the anisotropic converted waves instead of SV and SH, since the latter terms have no actual physical meaning for a vertically propagating wave. Their use is therefore confusing for non-specialists in RF. However, if they insist in using SV & SH for coherence with previous RF studies, as argued in the response to the reviewers, they should complete the statement added to the revised version by clearly stating that the studied waves have almost vertical propagation directions and, by consequence, SV and SH have not a real significance.

The reviewer is right that there is no physical meaning for a vertically travelling wave, but our waves propagate with an angle of 8-14 degrees from the vertical. This allows defining and discriminating between the two waves and they do have a physical meaning. Our use is according to the definition of SV and SH, which both oscillate perpendicular to the ray direction, with SV oscillating in the vertical plane and SH oscillating in the horizontal plane. We prefer to maintain this nomenclature because it is in accordance with the paper describing the software we use for calculating the synthetic receiver functions (Frederiksen and Bostock, 2000). We have already added a methods section describing the method and described the definition and we have added an explanatory sentence to avoid misunderstandings which now has been updated to clarify further.

Below are some minor comments that I hope will be useful in preparing the final version:

1- The verb is missing in the 1st phrase of the last paragraph of the results section. "which indicate the directions of the fast polarization axes (ϕ) in the lower and upper crust" ??? I guess that the missing verb is "differ"?

The sentence does not lack a verb: PSH-RF identifies two directions, "which indicate the directions of the fast polarization axes (ϕ) in the lower and upper crust".

2- Dykes are roughly vertical planar structures. If the basalts have a marked contrast in seismic velocity with the surrounding crustal rocks, a dyke swarm will produce anisotropy with a hexagonal symmetry with a slow axis normal to the dykes' trend. However a dyke swarm cannot produce the highly inclined symmetry axes of anisotropy proposed for most of the crust based on the RF data (6-40 km depth, cf. Table 1).

The reviewer is right that vertical dykes cannot by themselves produce anisotropy with an inclined axis. However, our argument is more complex. Pre-existing crustal fabric (with inclined fast axis) defines the overall anisotropy and weakness directions of the crust. Dykes tend to intrude along the weakness directions, i.e. parallel to the horizontal component of the fast axis. The combined effect of dyke swarm and pre-existing anisotropy will still have fast axis with the horizontal component parallel to the pre-existing anisotropy. However, the dip of the fast axis may be affected by the presence of the dykes while the total anisotropy increases by the addition of the dykes. We have added an explanation to make the argument clear.

3- Similarity in the fast polarization axes in the crust and the lithospheric mantle. I agree that if the two directions were markedly different between the crust and the mantle, the apparent SKS splitting should deviate from the fast direction in the crust. However, the strongest evidence for consistency of the fast directions along the SKS path would be absence of back-azimuthal variation in the observed fast polarization directions and delay times for the SKS. From what I remember the back-azimuthal coverage is not very good, but maybe for some stations this point can be tested?

Figure 2 shows the event distribution with an almost complete back-arc-coverage (cf. Fig.2 below). Therefore, the SKS estimates have a good coverage. The SKS results are not our interpretation and we use them for comparison with our crustal results to infer a connection between anisotropy in crust and upper mantle.

Reviewer #4 (Remarks to the Author):

As I was contacted by the editor to replace one of the reviewer for the second round, I will try not to cover the topics that have been already raised and I will not provide a very detailed review.

This paper addresses a difficult topic, the separation of crustal and mantle contribution to seismic anisotropy. Indeed the vertical resolution of body waves is always rather poor. The authors try to circumvent the problems by using Ps crustal conversions to isolate the crustal contribution. But as correctly pointed by one of the reviewer, it is sometimes difficult to separate the effect of anisotropy from those coming from dipping isotropic structures. The main issue is that it is also unclear what causes crustal anisotropy, in contrast to mantle anisotropy which presumably mainly comes from lattice-preferred orientation of olivine aggregates. Therefore the apparent fast direction in the crust may not necessarily be the principal shear direction as in the mantle. This caveat could perhaps be discussed.

Thank you. We have added to the discussion about these issues.

In addition to properly model the effects of seismic anisotropy on seismic waveforms one needs to consider the complete elasticity tensor. Indeed, as pointed out by the reviewers shear wave splitting and Ps conversions are not sensitive to the same elasticity coefficients so this has to be kept in mind when interpreting the results of modeling. Perhaps my main concern is that in any case the synthetic seismograms bear only a vague resemblance to the observed data. I think that quantifying the waveform misfit might be important to discriminate between the different models to try to convince the reader that indeed these models indeed fit the data and are robust.

We agree with the reviewer that it would be desirable to carry out a quantitative comparison between synthetic and observed receiver functions, but such comparison is not possible with current methods and data quality. The reviewer here repeats an earlier proposal from reviewer 2, who now accepts our argument that the proposed quantitative comparison is not technically feasible and R2 now recommends publication. Many authors have previously interpreted crustal anisotropy from Ps receiver functions by similar method to the one used in our paper. None of these papers includes a quantitative comparison between synthetic and observed receiver functions because it is technically impossible to measure the misfit between synthetic and observed sections due to the low signal to noise level and due to differences in the detailed traveltimes between observed and synthetic receiver functions. The list of authors on such papers includes Jeffrey Park, Yale (the authority on the use of receiver functions for determination of anisotropy) and Lev Vinnik, Russian Academy of Science (the founder of the receiver function method). The proposal to make such quantitative comparison would imply that we should develop a whole new method, whereas we have used state-of-the-art interpretation by visual comparison. We emphasise that we estimate the anisotropy by use of a standard method, which previously has been applied in young tectonic regions. The novelty in our paper is not the method, but the observation of strong seismic anisotropy in the cratonic crust, which has never before been observed.

Regarding the interpretation, the apparent coherence between crustal and mantle anisotropy could indeed suggest some vertically coherent deformation of the cratonic lithosphere of the Kaapvaal craton. Since the overall anisotropy is weak, this would suggest a very strong and hard to deform cratonic lithosphere, which raises the question as the rheology of this lithosphere at Archean times

with a much higher geotherm.

First, the measured anisotropy is not small but very substantial.

Second, the question of palaeo-geotherm and its implications for the formation of anisotropy in crust and mantle is interesting, but we prefer to abstain from entering this discussion because the available data needed to enter such discussion is not really available.

Finally, I think that the bibliography is uncomplete. I would like to see some more thorough comparisons with the results of surface wave azimuthal anisotropy studies in that region. There is also a nice paper by Fouch et al. (2004) which evidenced large small-scale variations of apparent splitting in the Kaapvaal craton. The results of this study demonstrate that there are also important lateral variations of seismic anisotropy in that region, and that stratification is not the complete story.

We agree and our results also show this. Reference to surface wave studies and the Fouch reference are now included.